# Implications of the COVID-19 Pandemic on the Organization of Remote Work in IT Companies: The Managers' Perspective

Michał Błaszczyk [1,*], Milan Popović [1,*], Karolina Zajdel [2] and Radosław Zajdel [1]

1   Department of Computer Science in Economics and Medicine, Faculty of Economics and Sociology, University of Lodz, 90-255 Lodz, Poland; radoslaw.zajdel@uni.lodz.pl
2   Department of Medical Informatics and Statistics, Medical University of Lodz, 90-419 Lodz, Poland; karolina@interforum.pl
*   Correspondence: michal.blaszczyk@uni.lodz.pl (M.B.); milan.popovic@uni.lodz.pl (M.P.)

**Abstract:** The study analyses the effects of the COVID-19 pandemic on work models and explores managerial perspectives on remote work compared to stationary work. A survey was conducted among companies, resulting in a sample of respondents. An additional research methodology used to validate the hypotheses was a market basket analysis. The findings indicate a significant change in work models, with a majority of companies adopting fully remote work or hybrid models with remote work as the predominant mode. Managers generally perceive remote work as having a significantly worse outcome compared to stationary work. Concerns about remote work include difficulties in supervising remote workers, maintaining effective communication with the team, and potential negative effects on employee motivation and well-being due to limited interaction. Preferences for work models varied, with a notable proportion favoring fully remote work or hybrid models. Reasons for considering a long-term change to stationary or remote work include coordination needs, control and supervision requirements, physical presence demands, and impacts on organizational culture and atmosphere. Benefits of remote work include employee flexibility, talent attraction, and alignment with employee demands and the labor market. This study provides insights into the evolving work landscape and informs strategies for effectively managing remote work environments.

**Keywords:** remote work; IT; managers; work organization; COVID-19

## 1. Introduction

Even though the time of the COVID-19 global emergency is over [1] and slightly becoming forgotten [2], as a consequence of the pandemic [3], a considerable proportion of Europeans engaged in remote work [3]. The influence of this change on the labor market is undeniably substantial [4–6]. The initial research [7] unveiled the widespread adoption of remote work and a notable change in the overall perception of remote employment. The increase in popularity of remote work, prompted by the COVID-19 pandemic and corresponding quarantine measures, necessitated its legal regulation [8]. Consequently, the Coronavirus Act was made into law to establish the framework for organizing remote work, addressing concerns related to employee health and safety, as well as data security requirements for remote work operations [9]. While some researchers initially equated remote work with telework [10], it is important to note that remote work, as defined in the Coronavirus Act, encompasses a broader scope than traditional teleworking, as per regulatory definitions [11]. The new legislation stipulates that, among other conditions, employers have the authority to instruct employees to carry out their designated tasks outside their usual place of work for a specified period, as a preventive measure against COVID-19 transmission [12]. The law does not specify the maximum duration of such arrangements or provide a clear definition of "work outside the place of its regular work". Moreover, the regulations do not explicitly outline the nature of remote work or address whether remote work can be initiated upon an employee's request.

Importantly, remote work does not mean complete exemption from supervisor surveillance. Employers retain the authority to provide ongoing instructions, assign specific tasks, and monitor the employee's performance during working hours. Measures such as requiring employees to remain reachable via phone or email during their designated work hours can be implemented to ensure work engagement [9].

The pandemic has amplified many of the factors highlighted in the literature as adverse outcomes of remote work. These encompass diminished interpersonal interactions with colleagues [13], blurring of boundaries between work and personal life [14], a persistent sense of being constantly engaged in work [13–16], conflicts between family and work obligations, and feelings of social isolation [17]. To establish a clear demarcation between work and personal life and safeguard worker well-being against the perils of an excessively interconnected work environment, it is imperative to implement specific measures, including the implementation of the right to disconnect [18]. While remote work is associated with these negative aspects, it also confers certain advantages such as cost and time savings, overcoming geographical barriers, elimination of office distractions [19], and increased flexibility [20]. Based on the 2020 data from the World Economic Forum, an overwhelming 80% of corporate leaders anticipated an adverse impact of remote and hybrid work on employee productivity, with approximately one in four foreseeing a substantial negative impact. One in six individuals believed it would yield no impact or even a positive effect on productivity [21]. Furthermore, the study on team virtuality revealed that both physical and informational social isolation were negative predictors of job satisfaction [22], which, in turn, exhibited a negative association with perceptions of remote work productivity [23]. On the other hand, certain research has highlighted how respondents perceived their own productivity to be enhanced during the COVID-19 pandemic, despite significant disruptions to their non-work life [24]. On the contrary, it has been emphasized that organizations aiming to optimize cost-efficiency for enhanced economic performance have pursued strategies to achieve financial savings [25]. In the context of globalization and organizational expansion, numerous challenges have emerged, including inadequate physical office space to accommodate employees and escalating energy expenses. To address these challenges and enhance competitiveness, enterprises seek increased flexibility [26], cost-effectiveness [27,28], and financial profitability [25].

## 2. The Initial Research [7]

The authors conducted a study to examine the impact of the COVID-19 pandemic on the labor market, specifically focusing on the IT sector in Poland. They found inconsistencies in existing studies and aimed to determine how remote work is perceived by IT employees in the country. The hypothesis was formulated that most IT sector employees in Poland perceive the transition to remote work positively and experience increased productivity.

The study reviewed the demand for IT professionals before and after the pandemic. It was noted that in the years leading up to the pandemic, there was stable demand for IT experts. However, following the announcement of the end of the pandemic, IT specialists became the most valued professionals, with a significant increase in job offers compared to pre-pandemic levels. Data for the study were collected through an online questionnaire distributed to IT sector employees. The questionnaire included questions about the participants' work model, self-evaluation of productivity, and whether they had changed employers. Demographic and employer-related data were also collected. The researchers used snowball sampling for data collection, acknowledging that it may not provide a representative sample.

To assess the representativeness of the sample, the obtained data were compared with a survey conducted by one of the leading job portals in Poland. Similarity indices and chi-square tests were employed to analyze the data and determine the level of concordance between the two surveys.

Overall, the study purposed to clarify the perceptions and experiences of IT sector employees in Poland regarding remote work during the COVID-19 pandemic.

The study examined the impact of remote work on productivity and employee perceptions in the IT sector in Poland and aimed to determine if the change in work mode to remote during the pandemic influenced productivity and employee perceptions. The survey revealed that a majority of respondents felt more productive when working remotely. Those who switched to remote or hybrid work models reported positive or no change in cooperation with colleagues, while a minority reported negative changes.

Other research studies from institutions such as Harvard University and New York University have also explored the impact of remote work during the pandemic, revealing interesting trends in electronic activity and communication. The results indicate that employers' concerns regarding the detrimental impact of remote work on productivity may be unfounded. However, observations regarding increased emailing outside of normal working hours raise concerns about the blurring of work-life boundaries.

The survey found that approximately one in three respondents preferred to work fully remotely, and a preference for a hybrid model with more remote work was also reported. A marginalized group preferred to work exclusively at the employer's location. Additionally, when asked if their employer forced a change in the work model, the majority of respondents who switched to remote, or hybrid work due to the pandemic stated they would consider changing employers.

In summary, the COVID-19 pandemic influenced employees' perceptions of their work models, with remote work being preferred by the majority of IT sector employees. These perceptions also influenced employee decisions regarding further employment with their current employer, highlighting the importance of the work model when choosing a new employer, particularly in a low unemployment context. Great emphasis has also been placed on the review of the existing subject literature. Therefore, in order to gain a comprehensive understanding of the subject literature, it is advisable to become familiar with primary research studies that address teleworking [10,29] and remote working [8,9,11,30,31] and their use during the COVID-19 pandemic [32–38], as well as flexible working conditions for remote working [14] and the intersection of private and professional life [15–17].

The compelling outcomes prompted the authors to undertake a comprehensive exploration of this phenomenon, adopting a viewpoint centered on IT enterprises, with a specific emphasis on managers responsible for supervising IT teams. Acknowledging the captivating nature of the results, the authors resolved to thoroughly investigate and analyze this phenomenon within the context of IT organizations.

## 3. Materials and Methods

Based on the findings above, consequently, the following opposite hypothesis can be formulated in order to bring them up for discussion, taking into consideration that digital transformation processes are being recognized as a major challenge for leadership and top management [39–42], particularly related to the COVID-19 pandemic [43–46]. The first hypothesis is that executives in IT companies will strive for a change in work mode to remote work, especially due to potential cost savings associated with remote work. The second is that executives in IT companies will strive for a change in work mode to on-site especially due to a decrease in employee productivity during remote work.

**Hypothesis (H1).** *Executives in IT companies will strive for a change in work mode to remote work, especially due to potential cost savings associated with remote work.*

**Hypothesis (H2).** *Executives in IT companies will strive for a change in work mode to on-site especially due to a decrease in employee productivity during remote work.*

To gather data from managers in the IT sector, the researchers created a web-based questionnaire. The decision to use an online format was made because the study aimed

to cover a wide geographical area. A total of 67 participants voluntarily accessed the questionnaire through a provided link sent via email to specific companies in the IT sector with the request to forward it to other IT managers. The researchers emphasized the anonymity and confidentiality of the research. Due to the online nature of the survey, the authors relied on snowball sampling for data collection [47], which limited their control over the selection of respondents. It is worth noting that snowball sampling does not provide a representative sample [48], and the authors acknowledged this limitation during the analysis.

The study encompassed a group of 67 participants, comprising individuals who held managerial roles within their respective organizations. This research aimed to comprehensively explore the multifaceted aspects of their attitudes toward changes with regard to remote work during and after the peak of the COVID-19 pandemic from IT managers' perspective. The primary focus of this research was to delve into the participants' perceptions, opinions, and behavioral patterns related to the concept of remote work. The authors designed a comprehensive survey, encompassing a range of variables that pertained to the post-pandemic work landscape.

Through a data collection process, the researchers probed the participants' sentiments regarding the viability, challenges, and benefits of remote work. Additionally, they aimed to determine the degree to which the respondents believed in the efficacy and productivity levels of their remote workforce. By gathering such valuable insights, the study aimed to provide an empirical foundation for understanding the complex interaction between managerial perspectives and the developing trends in remote work practices. It is essential to acknowledge that this analysis of the provided data may not be representative of the broader population or specific industries. The data provide information about changes in work arrangements and the reasons behind these changes.

The questionnaire comprised two sections. The first section inquired about participants' work arrangements, assessment of productivity, and whether there had been any changes in their employment status. The second section focused on collecting respondents' demographic information, as well as data on their companies. The survey was completely anonymous.

The additional research method used to verify the hypotheses was a technique called market basket analysis. This technique is used to detect associations, i.e., relationships or correlations between groups of elements that tend to occur together [49]. Associations are one of the six data mining models [50] classified under the category of non-pattern taxonomy models [51]. Considering its proven effectiveness, this methodology finds utility in numerous realms of research, encompassing customer preference analysis, support for human resource management, and even extending to investigations into the historical progression of language [52].

Market basket analysis enables the discovery of associations within datasets, allowing us to ascertain that, for instance, customers purchasing product A also buy products B, C, or D. Consequently, an association rule takes the form $X \rightarrow Y$, where $X$ and $Y$ are sets of attributes, with $X$ referred to as the antecedent and $Y$ as the consequent in this analysis [51]. In this context, when the variant of variable $X$ takes the value of 1, signifying true, or in the case of analyzing the availability of digital democracy tools, 0, signifying false, the variant of variable $Y$ will adopt the value of 1 with a certain probability. The analysis of market basket results can be interpreted using two coefficients: "confidence" and "support", which are expressed by the following equations:

$$confidence = P(X \mid Y) = \frac{(X \cap Y)}{X}$$

$$support = P\,(X \cap Y) = \frac{(X \cap Y)}{n}$$

where "*confidence*" refers to the conditional probability, indicating the likelihood of event *Y* happening when event *X* has occurred [51]. "*support*", on the other hand, represents the combined probability of both events occurring in the entire analyzed sample [51].

## 4. Results

The study carries out a statistical analysis of the obtained results to verify the hypothesis, as well as a market basket analysis as an additional research method.

The data indicate that the majority of the companies in which the participating managers were employed, 75% of them, have Polish capital or a majority of Polish capital. The other 25% of the companies have foreign capital or a majority of foreign capital. Among the provided job titles, manager has the highest count of 34 and it represents the largest cumulative percentage of 51% when considering all the job titles. Team leader follows with a count of 19 and a percentage of 28%. Director has a count of 10, representing 15% of the respondents. Owner, CEOs Board members, and CTOs have smaller counts and percentages compared to the other job titles, total 14%.

The data also provide us with insights into how long respondents have held managerial positions. A smaller proportion, 7%, consists of individuals with up to 2 years of experience; 22% of the respondents have 2 to 5 years of experience, while 43% have 5 to 10 years of experience. Furthermore, 27% of the respondents have more than 10 years of experience in managerial roles.

As a follow-up to the aforementioned study, the participants were initially asked to provide insights into their company's work model prior to the COVID-19 pandemic. The most common work arrangement among the provided options is fully on-site work, with a count of 37 and a percentage of 55%. This means that a majority of the individuals in the dataset had a work arrangement that required them to be physically present at a designated workplace. The second most common work arrangement is hybrid work with a predominance of on-site work, with a count of 15 and a percentage of 22%. This indicates that a significant portion of the individuals had a work setup that combines both on-site and remote work but with a higher emphasis on on-site work. Hybrid work with a predominance of remote work had a count of 9 and a percentage of 13%. This implies that a smaller proportion of individuals had a work arrangement that involves a combination of on-site and remote work but with a higher emphasis on remote work. Fully remote work has the lowest count of 6 and the lowest percentage of 9%. This shows that a minority of individuals in the dataset had a work arrangement that is completely remote, where they can perform their duties from any location without the need to be physically present at a specific workplace.

Continuing the research inquiry, the subsequent question posed to the participants was, "Has the work mode within your organization undergone any changes as a result of the COVID-19 pandemic?". Among the individuals in the dataset, the majority (63%) reported a change in their work arrangement due to the COVID-19 pandemic, transitioning to fully remote work. This indicates that a significant number of people had to change from working in a traditional office setting to working remotely from home or other locations. A notable proportion (28%) also reported a change to hybrid work with a predominance of remote work. A small portion (4%) reported a change to hybrid work with a predominance of on-site work. An equal number (4%) indicated that their work model did not undergo any changes during the pandemic.

Continuing with the survey responses, respondents were asked to answer the question if the work mode has changed or if such a change is planned due to the end of the COVID-19 pandemic emergency. The majority of respondents (49%) stated that there had been no changes in the work model. For those who reported changes, the most common transition was to a hybrid work model with a predominance of in-office work (21%), followed by a hybrid work model with a predominance of remote work (16%) and full-time remote work (13%). It is worth mentioning that no respondent declares changing work mode to fully stationary.

Subsequently, they were asked, if, after the end of the pandemic, a long-term change to work with a predominance of stationary work is made or planned, and what are the reasons for this decision. Among the respondents who provided an answer, most commonly cited reasons for such changes were the need for better control and supervision over employee work and the necessity for greater coordination and teamwork. Both reasons were selected by 30% of the respondents. Other significant factors influencing the decision included the need for greater physical presence in the office due to job nature or client interactions (24%), the impact on organizational culture and work atmosphere (19%), and difficulties in maintaining motivation, effective virtual communication, and work-life separation during remote work (ranging from 6% to 12%). A considerable portion of respondents (48%) either did not plan such a change or the question was not applicable to their situation. It is worth mentioning that problems with internet connectivity, technical requirements of remote work, or lack of adequate equipment or infrastructure for remote work had no impact on such decisions.

Similarly, respondents were asked, if after the end of the pandemic emergency, a long-term change to work with a predominance of remote work is made or planned, and what are the reasons for this decision. Among the respondents who provided an answer, the most frequently cited reasons for making or planning a long-term change in work model with a predominance of remote work were the need to adapt to the job market in response to employee demands (39%), greater flexibility for employees (34%), and the possibility to recruit employees from outside the local area (30%). Other factors influencing the decision included cost savings (18%), increased productivity and more efficient use of working time (13%), and the protection of employees' health and safety (4%). A significant portion of respondents (18%) either did not plan such a change or the question was not applicable to their situation.

The next issue that was covered concerned general observations on remote work, from the managers' perspective, compared to stationary work. The largest proportion (43%) perceived remote work to be slightly worse compared to on-site work. This suggests that they observed some challenges or limitations in the remote work setup. A significant percentage (27%) perceived remote work to be significantly better, indicating that they observed clear benefits or advantages in the remote work arrangement. Some managers (13%) reported that remote work was slightly better compared to on-site work. A smaller percentage mentioned that remote work was significantly worse or equal compared to on-site work. A few managers (9%) found it difficult to provide a clear assessment and chose the option "hard to say".

Respondents were also asked about their concerns about remote work in terms of efficiency and effectiveness. The most commonly cited concern was the lack of interaction and inability to establish interpersonal relationships may negatively impact employee motivation and well-being, as mentioned by 72% of respondents. Other significant concerns included difficulties in maintaining effective communication with the team with 49% of respondents and difficulties with supervising remote employees with 33% of respondents. Some of them also expressed concerns about limited project monitoring, distractions in the home environment, technical failures or equipment problems, and the lack of a dedicated workspace away from other household members. A small percentage of respondents (13%) indicated that they had no concerns regarding remote work.

Respondents were asked for their opinion on the benefits that were perceived in terms of costs associated with remote work compared to stationary work. The most commonly observed benefit of remote work in terms of cost savings was savings on commuting and business travel costs, mentioned by 72% of respondents, which resulted in a total of 32% of the given responses. Other significant cost-saving benefits included savings on office rental with 55% of respondents and 25% of all responses and savings on maintaining a permanent office infrastructure with 45% of respondents and 20% of responses. Some respondents also mentioned cost savings related to employee benefits, office supplies, and maintaining

office infrastructure. A small percentage of respondents indicated that they did not see any benefits in terms of cost savings associated with remote work.

An evaluation of the effectiveness of the tools and technologies used to monitor employees' remote working was also covered. Among the respondents who provided an evaluation, the most common assessment of the effectiveness was rather effective, mentioned by 31% of respondents. The next most frequent evaluation was moderately effective with 19% of respondents. A significant portion of respondents (40%) indicated that they either do not use or do not have experience with such tools, suggesting a lack of familiarity or utilization of monitoring technologies for remote work. A smaller percentage of respondents considered the tools and technologies to be very effective.

Managers were asked whether their companies had made any investments in tools and infrastructure supporting remote work. The majority (52%) indicated that their company has made such investments. A significant portion of respondents (27%) reported that their company has not made any investments in tools and infrastructure for remote work. A smaller percentage of respondents (21%) were uncertain or found it difficult to say. Consequently, managers were asked what steps have been taken in the company to ensure connectivity and communication between teams working remotely. Among the respondents who provided an answer, the most common action taken by companies to ensure connectivity and communication between remote teams was the utilization of online collaboration platforms such as MS Teams, Google Meet, or Slack, mentioned by 88% of respondents. Another significant measure was the use of video conferencing tools, cited by 73% of respondents. A small percentage of respondents reported that no specific steps were taken in their company to facilitate connectivity and communication between remote teams.

The next question asked the manager's opinion on the most suitable work model for their teams. The most preferred work model that best suited their teams was full-time remote work, with 43% of respondents choosing this option. The next most favored work model was the hybrid work model with a predominance of remote work, selected by 34% of respondents. A smaller portion of respondents preferred the hybrid work model with a predominance of on-site work, with 16% of respondents choosing this model. The least preferred work model was full-time in-office work, with only a few respondents selecting this option.

Managers were also asked how they rate employees' productivity during remote work compared to stationary work. There was a range of opinions regarding employees' productivity. A significant share of respondents (24%) found it difficult to make a direct comparison or did not have sufficient data to assess the productivity difference. Respondents felt that productivity was significantly higher during in-office work (13%) or moderately higher during in-office work (18%). On the other hand, respondents believed that productivity was significantly higher during remote work (18%) or moderately higher during remote work (27%).

The issue was also raised on the impact of the remote work model, which it has on cooperation with employees in the same team. The majority of respondents (51%) reported a positive impact of remote work on collaboration with team members. A significant portion (27%) expressed a negative impact, while a smaller percentage (13%) found it difficult to determine or had no comparison to make. A small proportion (9%) indicated that remote work had no impact on such collaboration.

The last question in the survey concerns the impact of the remote working model on cooperation with employees in the company in other teams. According to the respondents' perceptions, 39% of them reported a positive impact of remote work on collaboration with employees outside of their own team. A significant portion (21%) expressed a negative impact on collaboration, while 27% found it difficult to determine or had no comparison to make. A smaller proportion (13%) indicated that remote work had no impact on collaboration with employees outside of their team. It is important to note that these responses

represent the perceptions of the respondents and may vary based on individual experiences and specific work contexts. The full results have been presented in Table 1.

**Table 1.** Survey results summary.

| Question | Answer | R | % |
|---|---|---|---|
| What was your company's work model like before the COVID-19 pandemic? | Fully stationary work | 37 | 55% |
| | Hybrid work with the advantage of stationary work | 15 | 22% |
| | Hybrid work with the advantage of remote work | 9 | 13% |
| | Fully remote work | 6 | 9% |
| Has work model been changed due to the COVID-19 pandemic? | Yes, to fully stationary work mode | 0 | 0% |
| | Yes, to hybrid work with predominantly stationary work | 3 | 4% |
| | Yes, to hybrid work with predominance of remote work | 19 | 28% |
| | Yes, to fully remote work | 42 | 63% |
| | No, the model has not changed | 3 | 4% |
| Has work model changed or is such a change planned due to the end of the COVID-19 pandemic? | Yes, to fully stationary work mode | 0 | 0% |
| | Yes, for hybrid work with predominance of stationary work | 14 | 21% |
| | Yes, for hybrid work with predominance of remote work | 11 | 16% |
| | Yes, for fully remote work | 9 | 13% |
| | No, the model has not changed | 33 | 49% |
| If, after the end of the pandemic, a long-term change to work with a predominance of stationary work is made or planned, what are the reasons for this decision | The need for greater coordination and teamwork | 20 | 30% |
| | Difficulty in keeping employees motivated while working | 8 | 12% |
| | The need for better control and supervision of employees' work | 20 | 30% |
| | Difficulties in maintaining effective virtual communication | 4 | 6% |
| | Problems with internet connectivity or technical requirements of remote work | 0 | 0% |
| | Lack of adequate equipment or infrastructure for remote work | 0 | 0% |
| | Demand for greater physical presence in the office due to the nature of the work or interactions with clients | 16 | 24% |
| | Impact on organizational culture and work atmosphere | 13 | 19% |
| | Difficulty separating work and private life when working remotely | 8 | 12% |
| | No such change planned/not applicable | 32 | 48% |
| If, after the end of the pandemic, a long-term change to work with a predominance of remote work is made or planned, what are the reasons for this decision. | Cost savings | 15 | 22% |
| | Increased employee productivity and more efficient use of working hours | 9 | 13% |
| | Greater flexibility for employees | 23 | 34% |
| | Ability to attract employees from outside the local area | 20 | 30% |
| | The need to adapt to the labor market in response to employee demands | 26 | 39% |
| | Protection of employee health and safety | 3 | 4% |
| | Impact on environmental protection and sustainability | 0 | 0% |
| | No such change planned/not applicable | 12 | 18% |
| | Hard to say | 3 | 4% |

**Table 1.** *Cont.*

| Question | Answer | R | % |
|---|---|---|---|
| What are your general observations about remote work, from the manager's perspective, compared to stationary work? | Significantly better | 18 | 27% |
| | Slightly better | 9 | 13% |
| | No change | 2 | 3% |
| | Slightly worse | 3 | 4% |
| | Significantly worse | 29 | 43% |
| | Hard to say | 6 | 9% |
| What are your concerns about remote work in terms of efficiency and effective-ness? | Difficulty in supervising the work of remote workers | 22 | 33% |
| | Limited ability to monitor project progress | 9 | 13% |
| | Difficulties in maintaining effective communication with the team | 33 | 49% |
| | Lack of interaction and inability to establish inter-employee relationships can negatively impact employee motivation and well-being | 48 | 72% |
| | Difficulty focusing due to numerous distractions in the home environment | 25 | 37% |
| | Technical failures, network problems or malfunctioning equipment | 11 | 16% |
| | I have no concerns | 9 | 13% |
| What benefits do you see in terms of costs associated with remote work compared to stationary work? | Savings on office rent | 37 | 55% |
| | Savings on maintenance of fixed office infrastructure | 33 | 49% |
| | Savings on office supplies | 15 | 22% |
| | Savings on commuting and business travel costs | 48 | 72% |
| | Savings on employee benefits, such as meals | 14 | 21% |
| | I do not see such benefits | 3 | 4% |
| How do you assess the effectiveness of tools and technologies used to monitor remote work of employees? | Very effective | 6 | 9% |
| | Rather effective | 21 | 31% |
| | Moderately effective | 13 | 19% |
| | Not very effective | 0 | 0% |
| | I do not use/I have no experience with such tools. | 27 | 40% |
| Has your company made any investments in tools and infrastructure supporting remote work? | Yes | 35 | 52% |
| | No | 18 | 27% |
| | Difficult to say | 14 | 21% |
| What actions have been taken in the company to ensure connectivity and communication between teams working remotely? | Video conferencing tools | 49 | 73% |
| | Use of online collaboration platforms such as MS Teams, Google Meet, Slack, Zoom | 59 | 88% |
| | Communication via email | 28 | 42% |
| | None | 5 | 7% |
| Which work model, in your opinion, best suits your team? | Fully stationary work | 4 | 6% |
| | Hybrid work with a preponderance of stationary work | 11 | 16% |
| | Hybrid work with a preponderance of remote work | 23 | 34% |
| | Fully remote work | 29 | 43% |

**Table 1.** *Cont.*

| Question | Answer | R | % |
|---|---|---|---|
| How do you rate employee productivity during remote work compared to stationary work? | Definitely higher when working remotely | 12 | 18% |
| | Moderately higher during remote work | 18 | 27% |
| | Definitely higher during stationary work | 9 | 13% |
| | Moderately higher during stationary work | 12 | 18% |
| | Hard to say/I have no comparison | 16 | 24% |
| What impact do you think the remote work model has on cooperation with employees in your team? | Positive | 34 | 51% |
| | Negative | 18 | 27% |
| | None | 6 | 9% |
| | Hard to say/I have no comparison | 9 | 13% |
| What impact do you think the remote work model has on cooperation with employees outside your team? | Positive | 26 | 39% |
| | Negative | 14 | 21% |
| | None | 9 | 13% |
| | Hard to say/I have no comparison | 18 | 27% |

An additional research methodology used to validate the hypotheses mentioned in the "Materials and Methods" section was a technique known as market basket analysis. This analytical approach aims to discern associations or correlations between groups of elements that exhibit a frequent co-occurrence pattern [49].

Market basket analysis facilitates the identification of relationships within datasets, shedding light on customers' purchasing patterns, particularly regarding the propensity to acquire certain products together. The interpretation of market basket results is guided by two pivotal coefficients: "confidence" and "support" [51]. "Confidence" denotes the conditional probability of an event occurring given the occurrence of another event, while "support" quantifies the overall probability of both events transpiring within the entire analyzed sample. The following calculations were performed by leveraging the capabilities of the statistical software Statistica version 13.3 [53], which is widely recognized and employed in the scientific community for its robust analytical tools and advanced data processing capabilities.

In other terms, concerning the transition of work mode to reduce costs and increase employee productivity, the application of market basket analysis enables the evaluation of the impact of this transition on productivity enhancement and cost reduction, as perceived by the IT executives. The "confidence" coefficient, therefore, identifies the proportion of all respondents belonging to the first group (predecessor), whereas the "support" coefficient indicates the proportion of all participants from the first group whose responses align with those of the second group (successor). In this study, it was decided that results with confidence and support levels below 5% would be excluded from the analysis.

According to the findings of the study, when respondents reported significantly higher productivity when working remotely, there is a 14.5% confidence that the most suitable work model for them is fully remote work. Additionally, this association is supported by 100% of the respondents who experienced increased productivity during remote work. It was also demonstrated that when respondents indicated significantly higher productivity during stationary work, there is a 19% confidence that the most appropriate work model for them is a hybrid approach with a predominance of on-site work. This association is supported by 81% of the respondents who perceived increased productivity during on-site work. The selected results have been presented in Table 2.

**Table 2.** Productivity and preferred work mode.

| Predecessor | | Successor | Confidence | Support |
|---|---|---|---|---|
| Productivity significantly higher when working remotely | $\Longrightarrow$ | Preferred fully remote work | 14.5 | 100.0 |
| Productivity significantly higher during stationary work | $\Longrightarrow$ | Preferred hybrid work with predominance of stationary work | 18.8 | 81.3 |

Individuals who reported a notable increase in employee productivity during on-site work, in two-thirds of the cases, indicated that their respective companies have either already implemented or intend to implement a hybrid model with a predominant focus on remote work. Conversely, half of the respondents who perceived a moderate increase in employee productivity during remote work stated that their companies have either already implemented or plan to implement a fully remote work model. The remaining findings of this productivity analysis are presented in the table below. The most relevant results have been presented in Table 3.

**Table 3.** Productivity and current or planned work mode.

| Predecessor | | Successor | Confidence | Support |
|---|---|---|---|---|
| Productivity significantly higher when working stationary | $\Longrightarrow$ | Hybrid work with predominantly remote work | 9.0 | 66.7 |
| Productivity moderately higher when working remotely | $\Longrightarrow$ | Fully remote work | 13.4 | 50.0 |
| Productivity moderately higher while working stationary | $\Longrightarrow$ | Hybrid work with predominantly stationary work | 7.5 | 41.7 |
| Productivity moderately higher while working at a fixed location | $\Longrightarrow$ | Hybrid work with predominantly remote work | 10.4 | 58.3 |
| Productivity definitely higher when working remotely | $\Longrightarrow$ | Fully remote work | 13.4 | 75.0 |

As expected, respondents who expressed that a hybrid work model with a predominance of on-site work best fits their teams have, in a substantial majority of instances (73%), either currently implemented or are planning to adopt such a hybrid model. Likewise, individuals who stated that a fully remote work model best aligns with their teams have, in a significant majority of cases (69%), either already embraced or are in the process of adopting a fully remote work approach. Notably, even though a few managers (6%) identified an on-site work model as the best fit for their teams, all of these IT enterprises have either already implemented or are planning to adopt a remote work model. The most relevant results have been presented in Table 4.

Half of the respondents who evaluate remote work significantly better than on-site work have either already adopted or plan to adopt a hybrid work model with a predominance of remote work. The other half, however, have either already adopted or plan to adopt a fully remote work model. In no cases did individuals with slightly lower perceptions of remote work respond that their companies have adopted or are planning to adopt a fully on-site work model at the assumed minimum level of support and confidence. Conversely, nearly half of these respondents (48%) declared that their companies have adopted or are planning to adopt a hybrid work model with a predominance of on-site work. It is worth emphasizing that these same respondents, despite having slightly lower perceptions of remote work, responded that their companies have ultimately adopted or

are planning to adopt either a remote work model or a hybrid model with a predominance of remote work. The most relevant results have been presented in Table 5.

**Table 4.** Working model which fits best and current or planned work mode.

| Predecessor | | Successor | Confidence | Support |
|---|---|---|---|---|
| Hybrid work with mostly stationary work | ⟹ | Hybrid work with predominantly stationary work | 11.9 | 72.7 |
| Fully remote working | ⟹ | Hybrid work with predominantly remote work | 9.0 | 20.7 |
| Fully remote working | ⟹ | Fully remote work | 29.9 | 69.0 |
| Hybrid working with mostly remote working | ⟹ | Hybrid work with predominantly stationary work | 13.4 | 39.1 |
| Hybrid work with mostly remote working | ⟹ | Hybrid work with predominantly remote work | 16.4 | 47.8 |
| Fully stationary work | ⟹ | Hybrid work with predominantly remote work | 6.0 | 100.0 |

**Table 5.** Perceptions on remote work and current or planned work mode.

| Predecessor | | Successor | Confidence | Support |
|---|---|---|---|---|
| Perceptions on remote work much better | ⟹ | Hybrid work with predominantly remote work | 13.4 | 50.0 |
| Perceptions on remote work much better | ⟹ | Fully remote work | 13.4 | 50.0 |
| Perceptions on remote work slightly better | ⟹ | Fully remote work | 9.0 | 66.7 |
| Perceptions on remote work slightly worse | ⟹ | Hybrid work with predominantly stationary work | 20.9 | 48.3 |
| Perceptions on remote work slightly worse | ⟹ | Hybrid work with predominantly remote work | 13.4 | 31.0 |
| Perceptions on remote work slightly worse | ⟹ | Fully remote work | 9.0 | 20.7 |

## 5. Discussion

The provided data offers insights into the work models adopted by companies before, during, and potentially after the COVID-19 pandemic emergency. Prior to the pandemic, a majority of companies adhered to a fully stationary work model, indicating a conventional office-based setup. However, the pandemic prompted a significant transformation in work arrangements. Most companies swiftly transitioned to fully remote work, demonstrating a widespread adoption of remote work practices. Additionally, part of companies embraced a hybrid work model, with a preference for remote work.

Looking ahead, nearly half of the companies do not anticipate altering their work models, suggesting a potential return to pre-pandemic practices. Nevertheless, the other half is contemplating long-term changes. Specifically, 21% are considering a hybrid work model with a predominance of stationary work, emphasizing the value placed on in-person collaboration. Furthermore, 16% are planning a hybrid work model with a predominance of remote work, indicating the sustained popularity of remote work arrangements. An additional 13% of companies intend to transition to a fully remote work model, signaling a permanent change to remote work.

The reasons behind selecting a predominantly stationary work model include the necessity for enhanced coordination, teamwork, and effective supervision of employees' work. Moreover, the demand for physical presence in the office due to job nature or client interactions was a significant factor. Organizational culture and work atmosphere were also influential considerations. On the other hand, the decision to adopt a predominantly remote work model was driven by factors such as meeting employee demands, providing greater flexibility, attracting talent from outside the local area, cost savings, and potentially increased productivity. Health and safety concerns were mentioned but held less prominence in the decision-making process.

Remote work, from the manager's perspective, is predominantly perceived as significantly worse compared to stationary work, as indicated by almost half of the respondents. Concerns about the efficiency and effectiveness of remote work revolve around difficulties in supervising remote workers' tasks, maintaining effective communication within the team, negative impacts on employee motivation and well-being due to limited interaction and relationship-building opportunities, difficulty focusing amidst home environment distractions, technical failures, or network problems. A significant number of respondents identified cost benefits associated with remote work, including savings on office rent, maintenance of fixed office infrastructure, commuting and business travel costs. The assessment of tools and technologies used to monitor remote work varied, considering them rather effective, lacking experience or not utilizing such tools.

In terms of the preferred work model for teams, fully remote work was favored by the largest proportion of respondents, followed by hybrid work with a predominance of remote work. Hybrid work with a predominance of stationary work and fully stationary work were less popular choices. Regarding employee productivity, opinions varied, with almost half of respondents believing that remote work led to higher or moderately higher productivity compared to stationary work. However, almost a third found it hard to make a direct comparison or had no preference. The impact of the remote work model on cooperation with employees within the team was perceived as positive by the majority and negative by a smaller proportion.

These findings give a better understanding of how managers see things regarding remote work, highlighting concerns regarding supervision, communication, motivation, and technological monitoring. Further research in this area could provide deeper insights into addressing the challenges and maximizing the benefits of remote work from a managerial standpoint. They highlighted the investments made in remote work tools and infrastructure, the prevalence of video conferencing and collaboration platforms, and the preference for remote or hybrid work models. The impact on productivity and cooperation varied, emphasizing the need for further research and analysis in order to gain a comprehensive understanding of the remote work landscape.

Summarizing the discussion, the obtained data highlight the notable change toward remote work during the COVID-19 pandemic, with many companies contemplating a hybrid approach in the post-pandemic era. The selection of either a predominantly stationary or remote work model depends on various factors, including the nature of work, employee preferences, coordination requirements, and organizational culture. Further research and analysis in this domain could provide valuable insights into the evolving landscape of work models.

## 6. Conclusions

Based on the research data, the following conclusions can be drawn regarding remote work from a managerial perspective.

The first one is on productivity perceptions. The varying perceptions regarding productivity during remote work highlight the complex nature of assessing performance in remote settings. While some respondents believed remote work led to higher or moderately higher productivity, a significant proportion found it challenging to make a direct comparison or had no preference. This indicates the need for further investigation and understanding of

the factors influencing productivity in remote work environments. Through the way in which productivity is perceived, it is not possible to accept or reject the assumption raised, stating that executives in IT companies will strive for a change in work mode to on-site due to a decrease in employee productivity during remote work.

Cost saving was among the important factors driving the desire to change to or stay in a job with a preference for remote working, but more important factors were found. These include greater flexibility for employees, the ability to attract employees from outside the local area as well as the need to adapt to the labor market in response to employee demands. This means that, from the managers' perspective, responding to the demands of employees, who, due to the COVID-19 pandemic, have started to prefer remote working, is definitely more important than saving money.

Among the other conclusions that can be drawn from the study conducted, the first one concerns investment in tools and infrastructure. The fact that a majority of companies have made investments in tools and infrastructure supporting remote work indicates a recognition of the need to enable and optimize remote work environments. This suggests that organizations are proactive in adapting to the changing work landscape.

The second issue is technology utilization. The high utilization of video conferencing tools and online collaboration platforms emphasizes their crucial role in maintaining connectivity and facilitating effective communication between remote teams. This reliance on technology showcases its significance in enabling remote collaboration and bridging the gap between team members.

The third conclusion is based on work model preferences: The preference for fully remote work and hybrid models with a predominance of remote work suggests a growing acceptance and adoption of flexible work arrangements. This trend aligns with the increasing recognition of the benefits and feasibility of remote work.

The last issue concerns cooperation dynamics. The generally positive impact of remote work on cooperation within teams indicates that remote collaboration can be effective and successful. However, the reported negative effects by a substantial portion of respondents suggest that challenges in remote teamwork and collaboration exist. Similarly, positive perceptions of cooperation with employees outside the team indicate that remote work can facilitate effective interactions with external stakeholders.

Moreover, the results of the market basket analysis indicated that even when companies have managers who view on-site work as more productive and have slightly less favorable perceptions of remote work, there are no plans to introduce a fully on-site work model.

These detailed conclusions emphasize the importance of investing in remote work infrastructure, leveraging technology for remote collaboration, understanding the dynamics of productivity in remote settings, addressing challenges, and fostering effective cooperation in remote work environments. These insights can guide organizations in optimizing remote work practices and policies to enhance productivity and collaboration. Overall, these conclusions highlight the growing recognition of the importance of remote work, the critical role of technology in facilitating remote collaboration, and the need for further investigation into productivity and cooperation dynamics. These findings can inform organizations in effectively implementing and managing remote work strategies in the future.

**Supplementary Materials:** The following supporting information can be downloaded at: https: //www.mdpi.com/article/10.3390/su151512049/s1, File S1: The study questionnaire.

**Author Contributions:** Conceptualization: M.B., M.P., K.Z. and R.Z.; methodology: M.B. and M.P.; software: M.B. and M.P.; validation: M.B., M.P., K.Z. and R.Z.; formal analysis: M.B. and M.P.; investigation: M.B. and M.P.; resources: M.B. and M.P.; data curation: M.B. and M.P.; writing—original draft preparation: M.B. and M.P.; writing—review and editing: M.B. and M.P.; visualization: M.B. and M.P.; supervision: M.B. and M.P.; project administration: M.B. and M.P.; funding acquisition: R.Z. All authors have read and agreed to the published version of the manuscript.

**Funding:** This research received no external funding.

**Institutional Review Board Statement:** Ethical review and approval were waived for this study, due to ethical committee reviews are required only when biological material (biological, medical, chemical, physical) is collected or when scientific research interferes with the human psyche and our research covers areas that do not qualify for the ethics committee's consideration. The study was anonymous, and voluntary, without any compensation to the participants. Approval for the study was not required in accordance with the University of Lodz Research Ethics Committee (https://www.uni.lodz.pl/fileadmin/user_upload/RULES_OF_PROCEDURE_OF_THE_UL_RESEARCH_ETHICS_COMMITTEE.pdf, accessed on 27 May 2023).

**Informed Consent Statement:** Informed consent was obtained from all subjects involved in the study.

**Data Availability Statement:** Data are contained within the article or Supplementary Material.

**Conflicts of Interest:** The authors declare no conflict of interest.

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
