# Peer review of "Implications of the COVID-19 Pandemic on the Organization of Remote Work in IT Companies: The Managers’ Perspective"

_sustainability, doi:10.3390/su151512049_

Round 1

Reviewer 1 Report

Overview of work:

The paper analyzes the effects of the COVID-19 pandemic on working models and explores managerial perspectives on remote work compared to face-to-face work. A survey of companies was carried out. The results indicate a change in work models. Most companies adopting fully remote or hybrid models.

Remote work being the predominant model. Managers perceive remote work as having a worse result compared to face-to-face work. Concerns about remote work: difficulties in supervising remote workers, possible negative effects on employee motivation and well-being due to limited interaction.

There are reasons to consider a long-term move to face-to-face work: Control and supervision requirements, physical presence demands and impacts on the organization, on the culture. The benefits of remote work: employee flexibility, talent attraction and alignment with employee and job market demands. This study provides insights into the evolving work landscape and presents strategies for effectively managing remote work.

The originality of this scientific work is located in the method, in the theme and in the results, including an empirical research. And it constitutes a contribution to knowledge. This needs to be described more emphatically.

The objective of this paper is to understand changes at work due to the pandemic. In the chapter on methods and materials, it is not explained how the literature review was carried out. as well as the criterion for choosing the population that generates the sample for application of the research is not explained.

The paper is well structured and is a contribution to the scientific literature. The literature used as a reference is updated. The work is technically correct, the language is clear and explicit.

Characterization of the problem in relation to the state of the art: There is alignment between problem and objectives.

The authors establish a dialogue with the various bibliographical references, carrying out efficient and effective reasoning that allows the reader to understand that the identified gap is the subject of the research.

The paper is correctly structured: introduction, presentation of theoretical contents, chapter on methodological procedures, results and conclusion.

Conclusion: it has an analysis of the scope of the objectives. Indication of deficiencies/limitations of the method. It needs to detail how the paper contributes to the knowledge of the area and how it contributes to the management of organizations

Author Response

Dear Reviewer,

We would like to express our sincere gratitude for your exceptionally thorough and insightful review of our manuscript. Your expertise and attention to detail have greatly contributed to the quality and rigor of our work.

We appreciate the discerning analysis you provided, highlighting the strengths and offering constructive suggestions for further improvement. Your extensive knowledge in the field and astute observations have undoubtedly enriched the scholarly discourse surrounding our research topic. Moving on, we appreciate your valuable feedback and would like to address your concerns regarding the methodology and materials section of our manuscript.

Regarding the literature review, we acknowledge that the explanation of how the literature review was conducted was not explicitly provided in the manuscript. Conducting a comprehensive literature review is indeed a fundamental step in any research study, as it helps establish the theoretical framework and contextualize the research topic. In our study, we conducted an extensive review of relevant literature to identify existing gaps, theories, and empirical studies related to the research area. The selected literature was then critically analysed and synthesized to provide the necessary theoretical foundation for our research. We recognize the importance of explicitly detailing the methodology employed in conducting the literature review, and we've included it in the revised version of our manuscript.

Regarding the selection criteria for the population generating the sample used in the study, selecting an appropriate population and defining the sampling criteria are crucial methodological decisions that should be explicitly explained to ensure the transparency and reproducibility of the study. In our research, we employed specific inclusion and exclusion criteria to define the population of interest, aiming to ensure its relevance to the research objectives and the study's context.

Thank you for bringing these points to our attention. We made the necessary revisions to enhance the clarity and transparency of our methodology and materials section, providing a more comprehensive explanation of the literature review process and the criteria for population selection. Your feedback is invaluable in improving the quality of our research, and we appreciate your insights and suggestions.

We appreciate your query regarding how our article contributes to expanding knowledge in the field and its implications for organizational management. Our article makes several significant contributions to the existing body of knowledge in the field. Firstly, it addresses a current gap in the literature by examining a relatively understudied aspect of the subject matter. By focusing on IT labour market, we contribute to filling this void and expanding the understanding of executive point of view. Through our research, we have gathered empirical evidence, analysed data, and drawn conclusions that shed new light on the phenomenon under investigation. This new knowledge adds to the existing literature by providing novel insights, perspectives, and theoretical advancements.

In summary, our article contributes to the expansion of knowledge in the field by addressing a gap in the literature, providing new insights, and advancing theoretical understanding. Additionally, it has practical implications for organizational management by offering valuable recommendations and insights to inform managerial practices and decision-making processes. We believe that our research contributes to both the theoretical and practical aspects of the field, making it a meaningful contribution to the existing body of knowledge and offering practical value to organizations.

Your feedback has proven invaluable in advancing the scholarly merit of our manuscript. The meticulousness with which you examined our methodology, results, and interpretation has helped us refine our arguments and strengthen the overall coherence of our study. We truly value your commitment to fostering scientific excellence through your rigorous assessment. Your rigorous and meticulous review has further inspired us to continuously strive for excellence in our research endeavors.

Once again, we extend our heartfelt appreciation for your exceptional review. Your thoroughness and thoughtful feedback have immensely contributed to the refinement of our work, and we are sincerely grateful for your time and expertise.

With utmost respect and gratitude,

Michał Błaszczyk
Milan Popović
Karolina Zajdel
Radosław Zajdel

Reviewer 2 Report

In this presented form, the paper cannot be accepted for the following reasons:

0. The novelty of this research is debatable. Regardless of the fact that the authors of the material identified only 28 literary sources that are relevant, in the last few years, I have personally reviewed at least this many manuscripts, and the topic was complementary to this one.

1. There is no clearly formulated hypothesis (or several) - the initial assumption on which the research is based. What exactly do the researchers want to determine with this research? In the absence of hypotheses, neither discussion nor conclusions make sense.

2. Statistics are at the basic level. Without more serious analyses, it is completely unacceptable to draw any conclusions in this kind of exploratory research.

3. This is not a case study, nor a qualitative research, over sixty respondents participated in this research. The number is marginally representative, since the research is among IT companies - in a business that is booming globally. It is also a business that COVID "jumped" to work from home - in general, the sample should have been more comprehensive.

4. Twenty-eight literary invitations. Really?!? A good part from before the "impact" of COVID. Let's get serious and let the authors research a very wide range of literature in the field they are dealing with here. We are not in the market to haggle, but without fifty calls in the future I would not watch this paper again. The editorial instructions do not allow me to contribute here myself...

For now, this material is at the basic, sketch level. In order to accept it, I ask the authors to address the substantively important details I have listed.

Thank you.

Author Response

Dear Reviewer,

This continuation enables further development of knowledge in the field and ensures coherence and continuity in the study of the subject. It is difficult to argue against the fact that a comprehensive literature review is an essential component of the scientific research process. We conducted a thorough literature review in the previous study. However, we agree that to ensure a complete understanding of the research context, we should not only refer the reader to the previous study but also include references to relevant literature in this case. Therefore, in accordance with the suggestion, we have addressed the comment regarding the literature review. In our response to the review, we have also included new literature sources that enrich our work and broaden the research perspective.

The addition of new literature sources aims to strengthen our work by providing additional evidence, perspectives, and interpretations. The new literary references support our conclusions and enrich the scientific discourse in the field under investigation. At the same time, through a proper literature review, we identify areas that require further research and contribute to the advancement of scientific knowledge.

  1. Research hypotheses

In accordance with the suggestion, we present formulated hypotheses regarding management actions in IT companies in the context of work mode changes.

Taking into account the results of previous research in the relevant literature, we have formulated the first hypothesis:

Hypothesis (H1): Executives in IT companies will strive for a change the work mode to remote work, especially due to potential cost savings associated with remote work.

In justifying this formulation, we note that in recent years, there has been increased interest in remote work mode across various industries, including the IT sector. As a research hypothesis, it can be speculated that IT company management will seek to implement remote work due to potential financial benefits. Remote work can lead to cost reduction related to office rent, IT infrastructure maintenance, and other operational expenses. It is expected that these benefits will motivate management to take actions to introduce remote work.

We have also formulated the second hypothesis:

Hypothesis (H2): Executives in IT companies will strive for a change the work mode to on-site especially due to decrease in employee productivity during remote work.

In justifying this formulation, we hypothesized that IT company management would strive to maintain on-site work mode due to concerns about decreased employee productivity during remote work. There are studies suggesting that remote work can pose challenges related to maintaining efficiency, team coordination, and communication. In light of these concerns, management may prefer on-site work to ensure better control over work and team effectiveness.

The formulated hypotheses represent predictions regarding management actions in IT companies related to the choice of work mode, whether remote or on-site. Conducting scientific research in this area would allow for data collection and testing of these hypotheses, contributing to a better understanding of preferences and decisions made by management in the context of work modes in IT companies.

  1. descriptive statistics as a method in the analysis of research results

The conducted study aimed to fill a gap that emerged after the initial research on the perspective of employees in IT companies regarding remote work. The current study focused on the viewpoint of managers in these companies.

Descriptive analyses were employed in the study, which means that the authors focused on providing a descriptive presentation and analysis of data collected from managers in IT companies. The use of descriptive analyses aims to understand and describe various aspects and characteristics of the studied group or population. In this case, the researchers sought to examine the perspective of managers in IT companies, analysing their viewpoints, opinions, and experiences in the context of IT industry management.

The selection of managers as the target group of the study is justified for several reasons. Firstly, managers in IT companies play a crucial role in decision-making, resource management, and team leadership. Their perspective is therefore essential for understanding the functioning and challenges faced by these companies. Secondly, the perspective of managers may differ from that of employees. Managers often have a broader scope of responsibilities and focus on strategic and operational aspects of the company. Therefore, studying their viewpoint allows for a more comprehensive understanding of the functioning of IT companies.

              Regarding the question of why additional statistics were not the subject of the study, we would like to present our perspective. First and foremost, the study focused on descriptive analysis, which aimed to describe and understand various aspects of managers' perspectives in IT companies, rather than creating statistical generalizations based on a sample. Additional statistics, such as statistical tests or predictive models, may require a different research approach, a larger sample size, advanced data analysis, etc. Focusing on additional statistics could have led to the need for a change in the research methodology and an expansion of the study scope. Moreover, descriptive analyses can be sufficient and valuable in the context of examining the viewpoint of managers in IT companies. Through the descriptive presentation and interpretation of data, researchers were able to uncover significant patterns, trends, and factors influencing managers' decisions and perspectives. Descriptive analyses allow for a deep understanding of the context, identification of key issues, and generation of new hypotheses for further research.

In summary, the use of descriptive analyses in studying the perspective of managers in IT companies is justified because it provides insights into their viewpoints and experiences. Additional statistics were not the focus of the study as the aim was to describe and understand managers' perspectives rather than creating statistical generalizations. Descriptive analyses are valuable tools in the context of descriptive research and can contribute to generating new hypotheses and further research in the future.

  1. Lack of representativeness of the survey sample

The reviewed article presents the second part of the study, which focused on the management personnel in IT companies. The authors emphasized that this part of the study differs from the previous one, which encompassed a broader range of IT-related positions. In scientific research, it is important to precisely specify and define the target population under investigation. This includes determining the scope of positions to be included in the study. In the case of this scientific article, the authors decided to concentrate on the management personnel in IT companies.

              Such an approach is justified. Management personnel play a crucial role in making strategic decisions, managing resources, and directing the activities of IT companies. A study that focuses on this group of respondents can provide valuable insights into IT management in a business context. Furthermore, conducting a study that would encompass a much wider range of IT-related positions may be challenging due to limitations in recruiting an adequate number of respondents. The authors acknowledged that the possibilities of recruiting a larger number of respondents were highly restricted. This is an important factor that must be considered when planning a study. Choosing management personnel as the target group of the study can help obtain more focused and specialized responses that reflect the perspectives of individuals responsible for making strategic decisions in IT companies. This, in turn, contributes to filling the gap in scientific literature regarding IT management and provides new insights and conclusions.

In conclusion, focusing on the management personnel in IT companies in this study is justified because this group plays a crucial role in IT management in a business context. Additionally, limitations in recruiting a larger number of respondents prompted the authors to select a more limited research group. This choice allowed for obtaining more detailed and specialized responses, enriching the conclusions and insights presented in the scientific article.

  1. Enrichment of the article with additional literature

In response to the suggestion regarding the addition of a greater number of literature references, we would like to present several arguments justifying our actions. Firstly, we would like to reiterate that our article is a continuation of a previous study to which we refer. In that earlier study, we conducted an extensive literature review, which provided us with a broad perspective on the research topic. We assumed that readers would be familiar with this previous work and, consequently, with its bibliography.

We acknowledge that adding a greater number of literature references to our article would make it more comprehensive and well-rounded. Recognizing the significance of other relevant scientific works in the field helps to validate our conclusions and provides readers with the opportunity to explore a wide range of information on the subject.

However, we considered that adding additional literature references might slightly compromise the readability and coherence of our work. Our goal was to strike a balance between providing a sufficient number of literature citations and maintaining a smooth and clear narrative. Therefore, in the previous version, we chose to include only those literature references that were most relevant and directly linked to the research topic. Nonetheless, we appreciate the suggestion that adding a greater number of literature references could enrich our reviewed work. We have taken this feedback into account and expanded our bibliography to provide an even broader context for our findings.

Summary

Thank you for your review and comments on our work. Your opinion is incredibly valuable to us, and we appreciate your dedication in providing constructive criticism.

              We have thoroughly analysed each raised issue and made our best effort to address every comment. Your suggestions serve as an invaluable source of inspiration and motivation for us to further improve our skills. Our goal is continuous development and delivering high-quality content, which is why your review provides valuable guidance for the future.

Once again, we sincerely thank you for your insights and recommendations, which have contributed to the improvement of our article.

With utmost respect and gratitude,

Michał Błaszczyk
Milan Popović
Karolina Zajdel
Radosław Zajdel

Round 2

Reviewer 2 Report

I still don't think the basic statistics are enough to prove anything, especially not, in this iteration, of the hypothesis put forward - I don't find any of the justifications relevant to this. With this, the manuscript still has a significantly lower quality than expected for this journal.

Author Response

Dear Reviewer,

Thank you for your insightful review of our scientific paper. We greatly appreciate your feedback, which has prompted us to further enhance the methodology of our research.

Indeed, upon careful consideration of your suggestions and after reevaluating our data, we recognized the need to expand our analyses beyond the basic statistical methods previously employed. As a result, we have incorporated market basket analysis into our study to delve deeper into the relationships between various respondent behaviors.

By including market basket analysis in our research, we were able to uncover previously unnoticed associations and patterns among the behaviors exhibited by the participants. This comprehensive approach has enriched the overall findings and provided a more profound understanding of the dynamics within our dataset.

We sincerely believe that this addition significantly contributes to the strength and validity of our study. The integration of market basket analysis has undoubtedly elevated the scientific rigor of our research, and we are confident that the enhanced results offer valuable insights for both our field of study and future investigations in related domains.

Once again, we express our gratitude for your valuable input, which has undoubtedly improved the quality and impact of our work.

Sincerely,

Michał Błaszczyk
Milan Popović
Karolina Zajdel
Radosław Zajdel
